# Tourists' Willingness to Adopt AI in Hospitality—Assumption of Sustainability in Developing Countries

Tamara Gajić [1,2,*], Alireza Ranjbaran [3], Dragan Vukolić [2,4], Jovan Bugarčić [2], Ana Spasojević [5], Jelena Đorđević Boljanović [6], Duško Vujačić [7], Marija Mandarić [2], Marija Kostić [2], Dejan Sekulić [2], Marina Bugarčić [8], Bojana D. Drašković [8] and Sandra R. Rakić [9]

1. Geographical Institute "Jovan Cvijić", Serbian Academy of Sciences and Arts, 11000 Belgrade, Serbia
2. Faculty of Hotel Management and Tourism, University of Kragujevac, 36210 Vrnjačka Banja, Serbia; vukolicd@yahoo.com (D.V.); bugarcicjovan@gmail.com (J.B.); mmandaric@kg.ac.rs (M.M.); tebraceae@yahoo.com (M.K.); dejan.sekulic@kg.ac.rs (D.S.)
3. Faculty of Management, University of Tehran, Tehran 1411713114, Iran; alireza.ranjbaran@ut.ac.ir
4. Faculty of Tourism and Hotel Management, University of Business Studies, 78000 Banja Luka, Bosnia and Herzegovina
5. Faculty of Economics, University of Kragujevac, 34000 Kragujevac, Serbia; ana.spasojevic1985@gmail.com
6. Faculty of Organizational Studies EDUKA, University Business Academy, 11000 Belgrade, Serbia; jelenaboljanovic@vos.edu.rs
7. Faculty of Tourism and Hospitality, University of Montenegro, 85330 Kotor, Montenegro; dule.v@t-com.me
8. Faculty of Civil Engineering, University Union—Nikola Tesla, 11000 Belgrade, Serbia; mbugarcic@unionnikolatesla.edu.rs (M.B.); bdraskovic@unionnikolatesla.edu.rs (B.D.D.)
9. Faculty of Entrepreneurial Business and Real Estate Management, University Union—Nikola Tesla, 11000 Belgrade, Serbia; srakic@unionnikolatesla.edu.rs
* Correspondence: tamara.gajic.1977@gmail.com

**Abstract:** This study explores the impact of artificial intelligence (AI) on customer perceptions and behavior in restaurants, airline companies, and hotel sectors within the hospitality industry of Iran. The primary objective is to analyze how AI affects customer trust, brand engagement, electronic word-of-mouth (eWOM), and tourists' readiness to use AI technologies. Using a comparative analysis approach and surveys, this research tests hypotheses about the effects of artificial intelligence on various dimensions of customer interaction. The findings highlight significant relationships between the quality of artificial intelligence and customer engagement metrics, such as trust and brand loyalty, which are crucial for understanding and predicting customer behavior in response to technological advancements. This study lays the groundwork for theoretical assumptions about sustainability in these sectors in developing countries, providing a basis for future empirical research that could validate these assumptions and explore broader implications of AI integration in enhancing sustainable practices within the hospitality industry.

**Keywords:** AI; hospitality industry; developing countries; willingness to use AI; assumption of sustainability

## 1. Introduction

Artificial intelligence, as defined by Bowen and Morosan [1], is a technology that enables electronic devices to mimic human behavior. Services are often delivered using smartphones, chatbots, and other AI-based technologies [2,3]. One of the most significant advancements in this technological revolution is the integration of artificial intelligence devices into the travel and tourism industry [4–6]. These devices can offer travelers a wide range of services and experiences designed to enhance their journeys [7–9]. AI technologies are adopted for a variety of reasons, but some of them include cost savings brought on by efficiency and effectiveness as well as the accessibility of vast amounts of data [10,11].

This study aims to explore the relationships between AI information quality, AI system quality, trust, customer brand engagement, electronic word-of-mouth, and willingness to use AI, with a particular focus on the restaurant industry, airlines, and hotels in Iran, a developing country. The application of artificial intelligence in the hospitality sector offers numerous potential advantages, but its integration in developing countries, which are less open to foreign visitors, remains poorly explored. Most academic works focus on developed countries [9,12], thus creating a significant gap in research on the perception and use of artificial intelligence by tourists in developing regions [13]. This research gap is further exacerbated by the unique cultural, political, economic, and technological factors that influence the dynamics of artificial intelligence in these contexts [14–16]. Also, the role of artificial intelligence in promoting sustainable business practices in the hospitality industry, key to fostering environmentally friendly and socially responsible tourism, is not sufficiently covered in global research. Through this research, this study seeks to enrich the discourse on the role of artificial intelligence in advancing the practice of sustainable tourism in developing countries.

Tourism in Iran represents an increasingly significant industry, garnering attention due to its wealth of cultural heritage, diverse landscapes, including mountainous regions, deserts, lakes, and the Caspian Sea coastline, as well as historical landmarks [17]. Prior to the COVID-19 pandemic, Iran experienced a notable increase in visitor numbers, with approximately 9.1 million tourists in 2019 alone, but in 2022, there was a decline, attributed to the challenging recovery post-pandemic, with a total of 4.1 million tourists [18]. Iran is renowned for its rich cultural monuments, including ancient ruins, Islamic architectural gems, and UNESCO World Heritage sites such as Persepolis, Esfahan's bazaar, and the Masjid-e Jame in Isfahan [19]. Additionally, Iran serves as a significant pilgrimage destination for Shiite Muslims, particularly cities like Mashhad and Qom, housing sacred Shiite shrines and thus contributing to religious tourism [20]. The Iranian government actively invests in tourism infrastructure development, including hotel construction, roads, and airports nationwide, to support tourism growth [21]. However, despite its potential for further development, Iranian tourism faces challenges such as geopolitical instability, sanctions, and political tensions, which may affect the country's attractiveness to international tourists [22].

Our research uses the framework of the SOR model (stimulus–organization–response), which has a wide application in researching the connections between input (stimulus), process (organism), and output (response) [23]. By analyzing how AI devices serve as stimuli, influencing tourists' cognitive, emotional, and behavioral responses [24–29], we aim to unravel the intricacies of this evolving relationship. The research sought to achieve the following objectives:

- Explore how AI devices serve as stimuli that influence tourists' cognitive, emotional, and behavioral responses during their travel experiences in Iran.
- Utilize the stimulus–organization–response (SOR) model framework as a theoretical lens to analyze and interpret the complex dynamics between AI devices and tourists.
- Investigate the role of cultural context and technology readiness in shaping tourists' perceptions of AI devices in Iran.
- Provide insights into the challenges and opportunities that developing countries like Iran encounter when integrating AI into their tourism sector.
- Contribute to the limited theoretical and empirical research on AI's impact on sustainable hospitality, particularly in Iran. This study provides valuable insights for academia, tourism stakeholders, and policymakers, highlighting AI's role in promoting sustainable practices both locally and globally.

This study contributes to our understanding of how elements such as information quality, trust, brand loyalty, and eWOM influence the acceptance of AI-driven services in the hospitality industry. The study, which focuses on the Iranian setting, offers data with theoretical and practical implications, revealing the dynamic interaction between travelers and the application of AI in developing countries. Despite the challenges of

demonstrating sustainability within the hospitality sector, especially in countries where this sector occupies a marginal position, it is crucial to first understand and anticipate tourists' interest in the use of AI. After that, it is necessary to theoretically consider the potential application of AI in the context of tourism development, and later its contribution to potential sustainability. However, in developing countries, the aspect of sustainability currently allows only theoretical assumptions and not empirical predictions of sustainability. This approach requires a comprehensive analysis to identify possible ways through which AI can contribute to the sustainable development of the hospitality industry, taking into account the specific challenges faced by developing countries. Such a methodological approach not only facilitates theoretical understanding but also lays the foundation for future empirical research on the impact of artificial intelligence on sustainability in the hospitality sector. At this stage, due to the nascent state of AI integration and the complex variables affecting sustainability outcomes in developing countries, definitive predictions regarding sustainability impacts are not feasible; theoretical assumptions prevail. For this reason, the authors focused on researching tourists' willingness to adopt AI, which, in turn, would provide a basis for strengthening sustainability theories. This emphasis on measuring tourist receptivity to AI applications in the hospitality industry serves as a preliminary step toward understanding how the integration of AI could potentially align with sustainability goals.

## 2. Theoretical Framework and Hypotheses

AI is becoming increasingly prevalent in the hospitality industry, bringing innovations that significantly enhance service quality and operational efficiency [30,31]. Over the last two decades, it has been extensively introduced into the hotel sector to provide efficient service before, during, and after travel [32]. Technologies such as machine learning, natural language processing, and robotic process automation enable companies to analyze vast amounts of data and provide tailored services based on them [33]. AI algorithms allow hotels and travel agencies to suggest personalized excursions and activities to clients based on their previous preferences and behaviors [34]. In addition to service personalization, AI contributes to more efficient resource management and cost reduction through the automation of routine and time-consuming tasks. Tung and Au [35] developed a model exploring how the integration of AI into operational processes in the hospitality sector can significantly reduce operational costs and increase efficiency. They claim that this technology also helps improve the customer experience by offering faster and more accurate responses to guest inquiries, often through sophisticated support systems such as chatbots. Al Emadi et al. [36] presented a theoretical model explaining how AI-based customer support automation can enhance users' perception of service speed and quality in hospitality. The same authors also highlight that advanced analytics help managers better understand guest behavior trends and patterns, which can lead to better strategic decisions and increased customer satisfaction. Murphy et al. [37] developed a model demonstrating how deep data analysis can uncover new opportunities for service personalization and hospitality optimization. Through such personalized service and enhanced hospitality, AI emerges as a key factor transforming traditional approaches in these dynamic industries, opening doors to new opportunities for growth and innovation. Alt [2] theoretically elaborates on how AI, as a disruptive technology, can radically reshape management approaches in dynamic industries like hospitality, leading them toward more innovative and efficient operations.

Research by Bulchand-Gidumala et al. [38] emphasizes the crucial role of artificial intelligence in promoting hotel business sustainability, among other contributions to the transformation of hospitality. Their study demonstrates how AI enhances internal processes by using data to increase competitiveness and allows employees to personalize services, thus directly improving guest satisfaction. Importantly, their research also highlights how AI contributes to the sustainability of operations, helping hotels reduce resource consumption and optimize energy management. Additionally, AI facilitates the regulation

of legal and ethical aspects related to data usage. These findings clearly show that AI not only improves operational efficiency but also plays a key role in promoting sustainable practices in hotel management. Artificial intelligence brings significant advantages in terms of hotel business sustainability, as emphasized by Castelli et al. [39]. They argue that AI solutions are often cheaper, faster, and less error-prone compared to traditional methods involving human labor, directly contributing to more efficient resource management and reduced operational costs. Furthermore, they emphasize that AI has the ability to uncover complex patterns in extensive datasets, enabling hotels to better understand and optimize energy and other resource consumption. Such analysis and prediction of consumption can lead to a reduction in the ecological footprint and the implementation of more sustainable practices, which is crucial for modern hotel management.

Although there are relatively recent studies on the application of AI in hospitality, some authors, such as Saydam et al. [40], who explore the use of AI and robotics in hospitality, point out a significant lack of research regarding the impact of AI on improving business and sustainability in hospitality. The same authors argue that out of all 123 studies they encountered dealing with AI applications in hospitality, they primarily focus on theoretical foundations, even when it comes to the impact of AI on business sustainability. Earlier, authors Brock and von Wangenheim [41] also highlighted the limited research addressing specific business functions and strategic implications of AI, further emphasizing the need for deeper examination of this topic. Additionally, author Tuomi [42] suggests that despite widespread application in hospitality, AI has yet to fulfill its full potential, particularly in proving business sustainability, which remains merely a theoretical notion after all empirical research to date. Research by Um, Kim, and Chung [43] and Nam et al. [44] underscores the impact of AI, especially through smart chatbots and customer service platforms, on enhancing hotel guest satisfaction. These AI technologies tailor the guest experience and boost operational effectiveness, which is crucial for sustainable practices. The study by Majid et al. [45] employs content analysis as a methodological lens to investigate the application of artificial intelligence in hospitality, with a particular focus on sustainable tourism development. Although there are certain initiatives to introduce AI technologies into the sector, their implementation in the Indonesian tourism industry remains significantly low. The research results indicate the need to arouse interest among key stakeholders, including scientists and policymakers, to encourage the spread and innovation of AI technologies. A key research proposal is the identification of business models that would enable stakeholders in tourism to effectively adopt AI, thereby promoting the development of sustainable tourism in Indonesia.

Authors Vinuesa et al. [46] present contrasting views on the application of artificial intelligence (AI). They emphasize that the rapid development of artificial intelligence requires adequate regulatory oversight and insight to ensure sustainable development. Without such oversight, there is a risk of lack of transparency, security, and ethical standards in the use of AI technologies. This perspective highlights the need for a balanced approach in AI implementation, which simultaneously supports innovation while ensuring user and societal protection. However, all works are of a theoretical nature, where there is no confirmation of the results of empirical research that truly contributes to sustainability. Therefore, our manuscript also explores tourists' willingness to accept AI in hospitality, which is crucial knowledge for setting the theoretical basis for assumptions about possible business sustainability if the results prove significant and applicable in other research on a regional and broader level.

## 2.1. Artificial Intelligence Quality

The rapid advancement of technology is transforming global tourism [9,47]. AI-driven chatbots and intelligent navigation systems are key examples [5,48,49]. AI's impact spans various sectors, including tourism and hospitality, where it enhances information and system quality [11,50–52]. Machine learning algorithms enhance data accuracy and reliability, offering travelers personalized recommendations [53]. Real-time data analysis provides

up-to-date information on weather, flights, and traffic [54–57]. AI also enhances system quality by automating processes, increasing reliability, and ensuring data security [58,59]. The systems are designed to handle more users or visitors as their demand increases, without manual intervention. Additionally, these systems have the capability to learn from their operations and experiences, thereby enhancing their performance and efficiency over time automatically [60]. This is typically seen in systems that use machine learning or adaptive algorithms that adjust based on user interactions or other data inputs [61]. AI elevates information and system quality in hospitality and tourism, benefiting businesses with efficiency and security and enhancing travelers' experiences worldwide as AI continues to evolve [62,63].

*2.2. AI and Customer Brand Engagement*

Artificial intelligence is transforming how consumers behave and adopt technology within the hospitality industry. This change marks a significant strategic move towards more holistic management of consumer brands and technological advancements [2]. Essentially, AI helps businesses in this sector understand and anticipate customer needs better, which, in turn, influences how they integrate and utilize new technologies to improve guest experiences and operational efficiencies [64]. This innovation offers 24/7 task completion, cost savings, error reduction, and effective marketing through chatbots, significantly enhancing operational efficiency [65,66]. Additionally, AI's capability to analyze social media, predict trends, and measure customer satisfaction not only benefits businesses but also shapes the broader range of tourism [67,68]. In the realm of customer brand engagement, AI acts as a transformative force. It fosters personalized relationships, and the intersection of AI and brand engagement reveals significant implications for industry practices [4,69–71]. By personalizing interactions, AI caters to individual needs and analyzes vast amounts of customer data, enabling the creation of highly tailored content, recommendations, and experiences [72]. This level of personalization is supported by social robots, virtual agents, and AI-generated marketing content, which provide instant responses and utilize NLP and generative algorithms for adaptable content creation, thereby enhancing the customer experience [73,74].

Furthermore, predictive analytics play a crucial role in ensuring timely and relevant customer interactions, which enhances customer satisfaction and loyalty [75]. Recommender systems further amplify this effect by offering personalized product recommendations that boost customer engagement [76]. By integrating AI into brand interaction tactics, businesses not only enhance customer engagement and loyalty but also significantly contribute to the sustainability of the hospitality industry. This is achieved by optimizing resource management and reducing waste, making AI a pivotal element in today's data-driven marketing landscape and a key driver of sustainability in hospitality.

**H1a.** *Artificial intelligence information quality has an impact on customer brand engagement.*

According to this hypothesis, high-quality information provided by AI systems can increase brand engagement [77,78]. Brodie et al. [79] examined the dimensions of customer engagement and its role in fostering brand loyalty and advocacy. This is consistent with existing literature, highlighting the importance of providing accurate, relevant, and valuable information through AI to improve brand engagement.

**H1b.** *Artificial intelligence system quality has an impact on customer brand engagement.*

This hypothesis deals with the quality of AI systems but links it to customer brand engagement, similar to H1b. It implies that higher-quality AI systems might boost brand engagement. Customer engagement is emphasized as a factor in fostering brand loyalty and advocacy in research by Brodie et al. [79]. AI systems serve as a point of contact for consumer interactions in this scenario, and their effectiveness may have an impact on how engaged customers are with the business.

**H2a.** *Customer brand engagement has a direct impact on eWOM.*

According to this premise, when consumers are actively involved with a company, their positive engagement experiences may motivate them to spread their ideas and experiences via electronic word-of-mouth channels. Although the exact relationship between customer brand engagement and eWOM may vary based on the setting and industry, this hypothesis argues that there is a direct and positive correlation between the two, suggesting that higher levels of engagement are linked to increased eWOM activity [80].

**H2b.** *Customer brand engagement has a direct impact on willingness to use artificial intelligence.*

Even though the precise relationship between customer brand engagement and willingness to use AI may vary depending on the context and industry, this hypothesis suggests that there is a direct and positive connection between the two, implying that higher levels of engagement are associated with increased openness to using AI offered by the brand [81]. In order to provide evidence and insights into the relationship between various factors, researchers frequently test such hypotheses using empirical data and statistical analysis.

*2.3. AI and Trust*

Because it imitates human intelligence, artificial intelligence, which is frequently considered a subfield of computer science, is inextricably tied to trust [77]. According to Nadimpalli [59] and Yang et al. [82] it ranges from basic AI that can replace humans in specific situations to advanced versions like strong AI and super strong AI that have amazing capabilities. Understanding AI's effects on various facets of our existence requires trust in it. In order for customers to engage with brands and be willing to employ AI-driven solutions, trust in AI is essential [83,84]. When customers feel confident in the accuracy and value of the information these systems deliver, they are more willing to interact with businesses using AI. Customer engagement, electronic eWOM, and AI adoption are all impacted by trust in AI [85]. Customers' propensity to employ AI is greatly influenced by positive eWOM, which is motivated by trust [86,87]. According to van Doorn et al. [88], engaged customers are more likely to accept AI-powered services provided by their chosen companies, which highlights the importance of engagement in fostering acceptance of new technologies in the hospitality industry. This engagement is underpinned by strong customer brand engagement, characterized by loyalty and active interactions, that not only enhances experiences but also builds trust in AI [89,90]. Trust in AI serves as a crucial bridge between the technology and consumer acceptance [91], facilitating smoother interactions and greater reliance on AI solutions. As trust in AI increases, businesses can ensure higher usage rates of AI-driven services, which leads to more efficient operations, reduced waste, and an overall positive impact on sustainability [92]. By integrating trusted AI technologies, the hospitality industry can streamline processes and optimize resource utilization, crucial for maintaining an environmentally friendly and economically viable operation [93]. This interconnectedness of engagement, trust, and sustainability forms a virtuous cycle that drives the future of hospitality management.

**H3a.** *Artificial intelligence information quality has an impact on trust.*

This theory suggests that people's trust in AI hinges on the accuracy of the information it provides. In simpler terms, if AI systems consistently deliver reliable and quality information, people are more likely to trust them [94].

**H3b.** *Artificial intelligence system quality has an impact on trust.*

This hypothesis focuses on the overall quality of AI systems and suggests that the better the quality of the AI system itself (regardless of the information it provides), the higher the level of trust individuals will have in that system. In the context of AI, the quality

of the system, including its accuracy, reliability, and user-friendliness, can contribute to users' trust in the AI's recommendations and actions [94].

**H4a.** *Trust has a direct effect on eWOM.*

This hypothesis proposes that trust is a direct influencing factor on eWOM. In other words, when people trust a brand or AI system, they are more likely to engage in electronic word-of-mouth activities, such as recommending it to others or sharing their positive experiences online [95].

**H4b.** *Trust has a direct influence on willingness to use artificial intelligence.*

This hypothesis suggests that trust plays a direct role in influencing an individual's willingness to use AI. Trust serves as a foundation for individuals to feel comfortable with and have faith in AI systems, ultimately leading to their willingness to engage with these technologies [94,95].

*2.4. AI and eWOM*

Electronic word-of-mouth plays a crucial mediating role in the relationships between trust, customer brand engagement, and the willingness to use AI-driven technologies. Research by Hajli [96] underscores the significance of eWOM as a moderating variable, shaping consumers' perceptions and behaviors. Trust in AI, a pivotal factor in technology adoption, is subject to the influence of eWOM. Positive eWOM can enhance consumers' trust in AI technologies [97]. When individuals encounter favorable recommendations and reviews from their peers or online communities, their trust in AI is bolstered, facilitating greater willingness to use AI. Similarly, customer brand engagement, characterized by active interactions and loyalty, can be amplified through eWOM. Research by van Doorn et al. [88] suggests that positive eWOM can deepen customer engagement with brands. When customers engage in eWOM activities, sharing positive experiences related to AI-driven services or products, their brand engagement is reinforced. Moreover, eWOM can directly influence consumers' willingness to use AI. Positive recommendations and reviews shared through eWOM channels can serve as a powerful catalyst for AI adoption, especially when individuals perceive the source as credible and trustworthy [87]. This credibility not only encourages broader acceptance and use of AI technologies but also significantly contributes to sustainable business practices. When customers trust and adopt AI solutions, businesses can leverage these technologies to enhance operational efficiencies, reduce environmental impact, and provide services that meet customer needs with greater precision. In turn, this enhances the sustainability of operations, creating a cycle where trusted AI applications help drive environmentally and socially responsible innovations in the hospitality industry [98].

**H5.** *eWOM has a direct impact on willingness to use artificial intelligence.*

This theory contends that eWOM directly influences people's willingness to utilize AI. Additionally, when considered credible, eWOM has a direct impact on AI adoption and serves as a potent stimulant [87]. Businesses must put a premium on trust, brand engagement, and encouraging good eWOM given the mediating role that eWOM plays in promoting AI adoption [88,96].

**H6a.** *eWOM plays a mediating role of stimulus and organism element (trust) on willingness to use artificial intelligence.*

The information individuals receive (stimulus) and their level of trust in AI are linked by eWOM, which ultimately influences whether or not they are willing to employ AI. According to research by Hussain et al. [97], eWOM is crucial in influencing consumer

choice and acting as a mediator between recommendations and other external cues and people's perceptions and behaviors.

**H6b.** *eWOM plays a mediating role as a stimulus and organism element (customer brand engagement) on willingness to use artificial intelligence.*

This hypothesis suggests that eWOM acts as a mediator between external stimuli (such as recommendations or reviews) and the organism element (customer brand engagement), with the outcome being an increase in customer brand engagement itself [65]. The proposed research model with set hypotheses is shown in Figure 1.

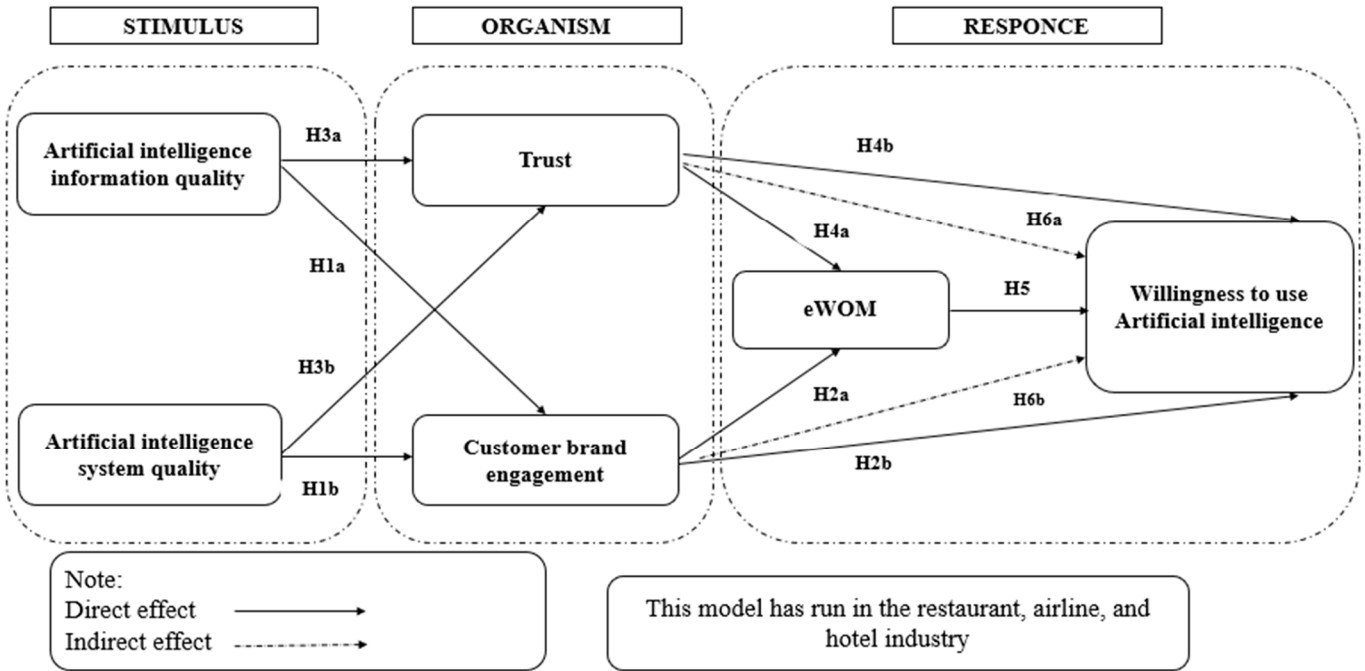

**Figure 1.** Conceptual framework of research.

### 3. Methodology

*3.1. Study Area and Participants*

This study was conducted on Kish Island, located in the Persian Gulf, within the nation of Iran, a developing country in South Asia. Kish Island is a prominent tourist destination, attracting over one million visitors annually due to its unique attractions such as Coral Beach, Kariz Underground City, Harireh Old City, and Horror Castle [99]. The island is also recognized for its free-trade-zone status, which enhances its appeal with local food restaurants, traditional resorts, and luxury hotels [99]. It boasts comprehensive connectivity through both domestic and international airports, serviced by more than 13 airlines offering direct flights. According to Camilleri [98], the hotel, restaurant, and airline industries play vital roles in any tourism destination. Projections indicate that the tourism and hospitality sectors on Kish Island will generate revenue exceeding USD 3102 million by 2023 [100].

This research utilized structured interviewing, which involved the use of predefined questions that were consistent for all participants, along with a specific sequence in which these questions were posed. Data collection involved distributing questionnaires to tourists utilizing various services on Kish Island, including hotels, restaurants, and airlines. This research was conducted from 2 July to 30 August 2023. Regarding questionnaire distribution, a total of 1200 questionnaires were distributed to tourists during this period. The distribution process was voluntary, allowing tourists the freedom to choose whether to participate in the survey or not. Participants were approached in hotels (246 respondents), restaurants (323 respondents), and at airport entrances (458 respondents). Surveyors explained the

questions to participants to ensure their understanding of the information being requested. This step was crucial, especially for complex questions, to enhance the reliability and validity of the responses. Out of the 1200 distributed questionnaires, 1027 were considered usable for analysis, resulting in a response rate of 85.33%. This response rate reflects the effectiveness of the distribution strategy and tourists' willingness to participate in the study. Table 1 presents the research results, indicating the sociodemographic characteristics of the research participants.

**Table 1.** Sociodemographic data.

| Characteristics | Relative Frequency | Frequency (%) | Relative Frequency | Frequency (%) | Relative Frequency | Frequency (%) |
|---|---|---|---|---|---|---|
| | Restaurant | | Airline | | Hotel | |
| Gender | | | | | | |
| Male | 175 | 49% | 159 | 48% | 181 | 53% |
| Female | 183 | 51% | 171 | 52% | 158 | 47% |
| Education | | | | | | |
| A–level and below | 18 | 5% | 15 | 4% | 22 | 6% |
| Under-graduate | 123 | 34% | 143 | 43% | 139 | 41% |
| Post-graduate | 185 | 52% | 152 | 46% | 168 | 49% |
| PhD | 32 | 9% | 20 | 7% | 10 | 4% |
| Age | | | | | | |
| 18–24 | 30 | 8% | 22 | 7% | 41 | 12% |
| 25–34 | 123 | 34% | 113 | 34% | 110 | 32% |
| 35–44 | 145 | 40% | 139 | 42% | 152 | 44% |
| 45–54 | 40 | 11% | 50 | 15% | 23 | 8% |
| 55 and above | 20 | 7% | 6 | 2% | 13 | 4% |
| Marital status | | | | | | |
| Single | 140 | 39% | 151 | 46% | 138 | 41% |
| married | 209 | 58% | 173 | 52% | 191 | 56% |
| divorced | 9 | 3% | 6 | 2% | 10 | 3% |
| Total | 358 | 100% | 330 | 100% | 339 | 100% |

The survey data indicate a relatively balanced gender distribution across the restaurants, airports, and hotel sectors, with a slight majority of females in restaurants and airports and a majority of males in hotels. Educational background shows that a significant portion of participants are highly educated, with post-graduates making up about half of the respondents in all sectors, followed by under-graduates. The age distribution leans heavily towards the 25–44 age group, which constitutes the majority in all sectors, suggesting that middle-aged adults are the primary consumers of hospitality services on the island. Marital status data reveal that married individuals are the predominant group in each sector, particularly in hotels, where they constitute 56% of the total.

*3.2. Research Method and Questionnaire Design*

The stimulus–organism–response (SOR) theory is a well-established model used extensively in the fields of marketing and tourism to analyze how environmental stimuli affect consumer behaviors [101]. According to this theory, an external stimulus triggers internal changes in an individual's mental state, which then lead to specific behavioral responses [102]. An "organism" in this context refers to the internal, psychological processes of an individual, encompassing cognitive (thoughts), emotional (feelings), and physiological (bodily) reactions that occur in response to a stimulus [103,104]. The theory categorizes behavioral responses into two types: approach behaviors, which are positive actions such

as exploring, staying, participating; or working and avoidance behaviors, which are characterized by hesitation or failure to interact constructively [105]. The "response" is thus the observable behavior resulting from these internal processes. The SOR model emphasizes the role of emotional or aesthetic elements of an environment in influencing these behaviors [106]. Building on this framework, Bittner [107] included cognitive and physiological aspects to expand the theory's application, particularly in the hospitality industry, integrating how cognitive and affective systems, along with past experiences and long-term memory, shape consumer responses to stimuli. This comprehensive approach helps in understanding the complex dynamics of consumer behavior in various settings [106].

The variables utilized in this study were sourced from previous literature and modified to suit our research contexts, specifically focusing on the hotel, restaurant, and airline industries. Adopting a Likert scale, we employed a 7-point measurement ranging from 1 ("strongly disagree") to 7 ("strongly agree") to assess the six factors with a total of 32 statements: artificial intelligence information quality (9 items) and artificial intelligence system quality [108] (6 items), trust [109] (5 items), customer brand engagement [109] (6 items), eWOM [75] (3 items), and willingness to use artificial intelligence [110] (6 items). These variables and corresponding questions can be found in Table 2. To ensure accurate translations without relying on mechanical means, we followed the recommendation of Ageeva et al. [111]. Initially, we distributed the questionnaire to expert academic lecturers in the field of tourism and hospitality who were originally from Iran but had been teaching in the United Kingdom and Australia for over ten years. Through this process, some items were modified and translated again to ensure the questionnaire was free from mechanical errors. The survey encompassed two parts. The pilot study involved the distribution of 25 questionnaires among professionals in the tourism and hospitality sectors. This step aimed to enhance the survey's quality and identify any necessary improvements including structures, the number of questions, and assessment of reliability. In the second phase, the questionnaire was administered to all travelers who had utilized services from the three industries mentioned earlier, namely, hotels (Aramis Plus, Vida, Dariush, and Maryam Sorinet), restaurants (Kooh Noor, Foodland, Mirmohana), and airlines (Iran Air, Mahan), which are renowned in the tourism and hospitality sectors in Iran according to Tripadvisor's latest reviews [112]. This questionnaire is composed of three parts, including a description of artificial intelligence and its productive role in the tourism and hospitality sectors. In the next part, we asked them to answer the following questions, such as their demographic information, and the main part of this questionnaire is about AI information quality, AI system reliability, trust, electronic eWOM, consumer brand engagement, and willingness to use AI.

**Table 2.** Statistical measures of construct validity and reliability for constructs.

| Items | Restaurants | | Airlines | | Hotels | |
|---|---|---|---|---|---|---|
| KMO Test | 0.869 | | 0.916 | | 0.826 | |
| Bartlett's Test ($X^2$) | 5158.8 | | 4567.1 | | 3546.9 | |
| df | 59 | | 55 | | 45 | |
| *p* | 0.012 | | 0.021 | | 0.019 | |
| Amount of ($R^2/Q^2$) | | | | | | |
| Constructs | $R^2$ | $Q^2$ | $R^2$ | $Q^2$ | $R^2$ | $Q^2$ |
| Trust | 0.18 | 0.06 | 0.396 | 0.152 | 0.368 | 0.202 |
| eWOM | 0.126 | 0.023 | 0.222 | 0.148 | 0.286 | 0.216 |
| Customer brand engagement | 0.241 | 0.125 | 0.326 | 0.226 | 0.357 | 0.236 |
| Willingness to Use AI | 0.141 | 0.075 | 0.265 | 0.096 | 0.247 | 0.148 |

*3.3. Data Analysis*

Prior to data analysis, a non-response bias test was administered by excluding the initial and final 50 responses for each variable. The results revealed no significant disparity between the first and last responses, thereby indicating the absence of non-response bias in the current study. In accordance with recommendations of authors Hair et al. [113], a two-step approach was adopted. The first step involved conducting exploratory factor analysis (EFA), using SPSS 22.00 software to evaluate the observed and latent variables while examining the relationships between them. EFA is a valuable technique for assessing internal reliability in cases where variables have not been previously examined [114]. The EFA results demonstrated the satisfactory reliability of the research components (Table 2). To determine the suitability of the data for factor analysis, we conducted the Kaiser–Meyer–Olkin (KMO) [115] test, which yielded a value of 0.869, 0.916, and 0.826 for restaurant, airline, and hotel industries, respectively (Table 2). This value exceeds the minimum acceptable threshold of 0.6, indicating that exploratory factor analysis (EFA) could be confidently performed on the data [116]. For the evaluation of our research hypotheses, we utilized SmartPLS 3 software, employing both confirmatory factor analysis (CFA) and structural equation modeling (SEM) [115]. These analyses allowed us to assess the relationships between variables and test our proposed hypotheses. Within this framework, we examined the $R^2$ values, which measure the proportion of variance in the endogenous variable explained by the exogenous variable(s). The results of these analyses are presented in Table 2. It is noteworthy that previous studies have recommended $R^2$ values to be greater than 0. Additionally, we also considered the $Q^2$ values, which indicate the predictive relevance of specific dependent constructs. Similar to $R^2$, $Q^2$ values are expected to be greater than 0 [116]. Both AI information quality and AI system quality variables, because of their roles as independent variables, do not have $R^2$ and $Q^2$ indices. In the subsequent section, we present the results of hypothesis testing, which will shed light on the relationships and outcomes of our study.

## 4. Results

*4.1. Results of Construct Reliability and Validity*

Table 3 presents a comprehensive analysis of how customers perceive AI tools across restaurants, airlines, and hotels, focusing on factors like AI information and system quality, trust in the service providers, electronic word-of-mouth, customer brand engagement, and willingness to use AI.

Customers generally view the information produced by AI tools as accurate and up to date, with statistical measures showing high means and strong reliability across sectors. This indicates a consistent and positive reception of AI's information capabilities. Furthermore, AI systems are perceived as reliable and adaptable, which supports their ongoing use in these industries. Trust is notably high, with customers believing in the commitment of these businesses to uphold promises, particularly regarding sustainable practices. This trust extends to customers actively recommending these brands online, suggesting a strong electronic word-of-mouth presence. Engagement levels are similarly high, showing that customers feel a deep connection and loyalty towards these brands, likely influenced by their positive interactions with AI technologies. Additionally, there is a marked willingness among customers to interact with and utilize AI services, reflecting an embrace of technological advancements in everyday service encounters. These collective data underscore a significant approval and readiness for AI integration in service industries, highlighting its role in enhancing customer experiences and operational sustainability.

**Table 3.** Measurement items, descriptive statistics, and reliability analysis for constructs.

| Items | Restaurants | | | | Airlines | | | | Hotels | | | |
|---|---|---|---|---|---|---|---|---|---|---|---|---|
| | FA | M | Std. | α | FA | M | Std. | α | FA | M | Std. | α |
| **AI information quality** | | | | | | | | | | | | |
| Tools of artificial intelligence (AI) produce accurate information | 0.753 | | | | 0.789 | | | | 0.832 | | | |
| There are a few errors in the information I receive from AI tools | 0.851 | | | | 0.748 | | | | 0.852 | | | |
| AI tools provide me with the latest information | 0.832 | | | | 0.729 | | | | 0.816 | | | |
| AI tools produce real-time information | 0.759 | 4.99 | 1.13 | 0.789 | 0.823 | 5.13 | 1.01 | 0.768 | 0.819 | 5.06 | 1.08 | 0.823 |
| The information from AI tools is always up to date | 0.789 | | | | 0.864 | | | | 0.845 | | | |
| The quality of AI systems potentially contributes to the sustainability of the hospitality industry by enabling reliable resource management | 0.825 | | | | 0.778 | | | | 0.869 | | | |
| **AI system quality** | | | | | | | | | | | | |
| AI tools operate reliably | 0.689 | | | | 0.784 | | | | 0.729 | | | |
| AI tools perform reliably | 0.798 | | | | 0.745 | | | | 0.789 | | | |
| The operation of AI tools is reliable | 0.789 | | | | 0.789 | | | | 0.746 | | | |
| AI tools can be adapted to meet various needs and potentially contribute to the flexibility of sustainable practices | 0.799 | | | | 0.741 | | | | 0.698 | | | |
| AI tools flexibly adjust to new requirements or conditions | 0.836 | 4.83 | 0.98 | 0.802 | 0.851 | 4.98 | 1.05 | 0.823 | 0.813 | 5.23 | 1.28 | 0.899 |
| AI tools are versatile in addressing needs as they arise | 0.821 | | | | 0.853 | | | | 0.827 | | | |
| AI tools respond quickly to my requests, which can contribute to sustainability | 0.835 | | | | 0.836 | | | | 0.752 | | | |
| AI tools provide information instantly | 0.854 | | | | 0.891 | | | | 0.749 | | | |
| AI tools quickly return responses to requests | 0.843 | | | | 0.786 | | | | 0.841 | | | |
| **Trust** | | | | | | | | | | | | |
| I trust this airline/hotel/restaurant (I believe) | 0.788 | | | | 0.812 | | | | 0.836 | | | |
| I believe this airline/hotel/restaurant is committed to keeping its promise to customers | 0.798 | | | | 0.826 | | | | 0.891 | | | |
| I believe this airline/hotel/restaurant is reliable for its customers | 0.795 | 5.01 | 1.03 | 0.795 | 0.829 | 4.89 | 0.970 | 0.801 | 0.798 | 5.06 | 1.06 | 0.826 |
| I would like this company to continue providing quality services with sustainable practices | 0.803 | | | | 0.874 | | | | 0.769 | | | |
| This airline/hotel/restaurant meets my expectations in terms of sustainability | 0.863 | | | | 0.865 | | | | 0.794 | | | |
| **eWOM** | | | | | | | | | | | | |
| I have recommended this brand in online pages to lots of people | 0.865 | | | | 0.834 | | | | 0.863 | | | |
| I "talk-up" the brand in online pages to my friends | 0.859 | 4.87 | 1.23 | 0.784 | 0.798 | 4.50 | 0.990 | 0.812 | 0.851 | 5.22 | 1.31 | 0.855 |
| I give this brand in online pages lots of positive word of mouth (advertising) | 0.789 | | | | 0.783 | | | | 0.825 | | | |
| **Customer brand engagement** | | | | | | | | | | | | |
| I am enthusiastic towards the brand | 0.698 | | | | 0.745 | | | | 0.76 | | | |
| I am passionate about the brand | 0.798 | | | | 0.769 | | | | 0.749 | | | |
| I have a sense of belonging to the brand | 0.789 | | | | 0.777 | | | | 0.812 | | | |
| When dealing with the brand, I am deeply engrossed | 0.865 | 5.11 | 1.05 | 0.826 | 0.768 | 5.16 | 1.25 | 0.826 | 0.827 | 5.45 | 1.26 | 0.859 |
| I am fully focused on how the brand manages its resources sustainably | 0.845 | | | | 0.836 | | | | 0.829 | | | |
| When I am engaged with the brand, my mind is focused on innovations and practices that enhance sustainability | 0.823 | | | | 0.815 | | | | 0.785 | | | |
| **Willingness to use AL** | | | | | | | | | | | | |
| I am willing to receive AI device airline/hotel/restaurant services | 0.897 | | | | 0.902 | | | | 0.874 | | | |
| I will feel happy to interact with AI devices in airline/hotel/restaurant services | 0.878 | 5.56 | 1.32 | 0.865 | 0.899 | 5.55 | 1.26 | 0.854 | 0.765 | 5.28 | 1.15 | 0.874 |
| I am likely to interact with AI devices in airline/hotel/restaurant services | 0.759 | | | | 0.877 | | | | 0.712 | | | |

Note: FA = factor loading; M = mean; Std. = standard deviation; α = Cronbach's alpha.

The discriminant validity test was conducted, and all average variance extracted (AVE) values were found to be greater than 0.5. This indicates that each research construct effectively measures a distinct variable. AVE values exceeding 0.5 are considered sufficient for demonstrating discriminant validity [113]. The discriminant validity index, which is smaller than 0.92, confirms the distinction between the variables, further supporting the validity of the research constructs (Table 4).

**Table 4.** Composite reliability and validity metrics for factors.

| Item | Restaurants | | | | | | | | Airlines | | | | | | | | Hotels | | | | | | | |
|---|---|---|---|---|---|---|---|---|---|---|---|---|---|---|---|---|---|---|---|---|---|---|---|---|
| | CR | AVE | IQ | SQ | T | eW | CBE | WU | CR | AVE | IQ | SQ | T | eW | CBE | WU | CR | AVE | IQ | SQ | T | eW | CBE | WU |
| IQ | 0.899 | 0.643 | 0.801 | | | | | | 0.859 | 0.623 | 0.789 | | | | | | 0.968 | 0.703 | 0.838 | | | | | |
| SQ | 0.956 | 0.653 | 0.523 | 0.808 | | | | | 0.965 | 0.655 | 0.689 | 0.809 | | | | | 0.854 | 0.597 | 0.685 | 0.772 | | | | |
| T | 0.854 | 0.655 | 0.256 | 0.457 | 0.809 | | | | 0.923 | 0.708 | 0.654 | 0.541 | 0.841 | | | | 0.958 | 0.67 | 0.596 | 0.659 | 0.818 | | | |
| eW | 0.863 | 0.702 | 0.451 | 0.569 | 0.369 | 0.837 | | | 0.953 | 0.648 | 0.452 | 0.536 | 0.706 | 0.804 | | | 0.968 | 0.716 | 0.589 | 0.665 | 0.642 | 0.846 | | |
| CBE | 0.889 | 0.647 | 0.325 | 0.596 | 0.568 | 0.754 | 0.804 | | 0.936 | 0.617 | 0.485 | 0.566 | 0.695 | 0.589 | 0.785 | | 0.974 | 0.63 | 0.639 | 0.623 | 0.635 | 0.458 | 0.793 | |
| WU | 0.895 | 0.717 | 0.369 | 0.512 | 0.589 | 0.695 | 0.658 | 0.846 | 0.956 | 0.796 | 0.562 | 0.736 | 0.655 | 0.532 | 0.458 | 0.892 | 0.923 | 0.618 | 0.725 | 0.689 | 0.569 | 0.569 | 0.685 | 0.786 |

Note: IQ = AI information quality; SQ = AI system quality; T = trust; eW = eWOM; CBE = customer brand engagement; WU = willingness to use AI.

Moreover, to assess the discriminant validity between the reflective constructs in this study, the heterotrait–monotrait (HTMT) test, introduced by Hew et al. [114], was employed. The HTMT test results, presented in Table 5, indicate that discriminant validity has been successfully established between the two reflective constructs.

**Table 5.** HTMT ratio analysis for AI constructs.

| Constructs | Restaurants | | | | | Airlines | | | | | Hotels | | | | |
|---|---|---|---|---|---|---|---|---|---|---|---|---|---|---|---|
| | WU | IQ | SQ | T | eW | WU | IQ | SQ | T | eW | WU | IQ | SQ | T | eW |
| IQ | 0.526 | | | | | 0.623 | | | | | 0.698 | | | | |
| SQ | 0.459 | 0.548 | | | | 0.648 | 0.687 | | | | 0.645 | 0.597 | | | |
| T | 0.548 | 0.578 | 0.542 | | | 0.589 | 0.712 | 0.452 | | | 0.587 | 0.526 | 0.478 | | |
| eW | 0.536 | 0.623 | 0.568 | 0.743 | | 0.542 | 0.562 | 0.396 | 0.268 | | 0.459 | 0.489 | 0.493 | 0.687 | |
| CBE | 0.651 | 0.653 | 0.423 | 0.765 | 0.689 | 0.621 | 0.459 | 0.354 | 0.369 | 0.529 | 0.487 | 0.598 | 0.562 | 0.651 | 0.751 |

Note: IQ = AI information quality; SQ = AI system quality; T = trust; eW = eWOM; CBE = customer brand engagement; WU = willingness to use AI.

Additionally, to examine the potential influence of factors, such as gender, education, age, and position on the research outcomes, hierarchical linear modeling (HLM) was utilized. The results of the HLM analysis revealed that none of these control variables had a substantial impact on the research outcomes. This is evident from the insignificance of the Beta's for the control variables, as their analysis results exceeded the threshold of 0.05 [113]. Thus, it can be concluded that the control variables did not exhibit any significant effects in this particular study.

*4.2. Hypothesis Testing*

The research results presented in Table 6 pertain to the examination of various hypotheses regarding the quality of information provided by AI, the quality of AI systems, trust, brand engagement among consumers, and eWOM in three industries: restaurants, airlines, and hotels. It has been demonstrated that the quality of AI systems and information plays a fundamental role in shaping trust and brand engagement, which further influences consumer behavior such as eWOM and willingness to use AI. Specific challenges and opportunities vary among industries, emphasizing the need for industry-specific approaches in the implementation and management of AI technologies.

AI information quality

It has been proven that trust in the information provided by AI positively influences consumers in all examined industries (Beta values are $\beta = 0.165$, $\beta = 0.398$, and $\beta = 0.120$, with *p*-values of $p = 0.023$, $p = 0.000$, and $p = 0.023$, respectively), confirming hypothesis H1a. As for H1b, concerning brand engagement among consumers, the hypothesis is confirmed in the restaurant ($\beta = 0.242$; $p = 0.001$) and airline industries ($\beta = 0.403$; $p = 0.000$), but not confirmed in the hotel industry ($\beta = 0.095$; $p = 0.148$).

**Table 6.** Hypothesis testing results for AI attributes.

| Hypothesizes Path | | | | Restaurants | | | | Airlines | | | | Hotels | | | |
|---|---|---|---|---|---|---|---|---|---|---|---|---|---|---|---|
| | | | | $\beta$ | *t*-Value | *p*-Value | Result | $\beta$ | *t*-Value | *p*-Value | Result | $\beta$ | *t*-Value | *p*-Value | Result |
| AI information quality | H1a | $\mapsto$ | Trust | 0.165 | 4.775 | 0.023 | Sig | 0.398 | 8.475 | 0.000 | Sig | 0.120 | 2.112 | 0.023 | Sig |
| | H1b | $\mapsto$ | Customer brand engagement | 0.242 | 3.043 | 0.001 | Sig | 0.403 | 8.610 | 0.000 | Sig | 0.095 | 1.696 | 0.148 | Not |
| AI system quality | H2a | $\mapsto$ | Trust | 0.447 | 9.378 | 0.000 | Sig | 0.227 | 4.638 | 0.012 | Sig | 0.446 | 9.715 | 0.000 | Sig |
| | H2b | $\mapsto$ | Customer brand engagement | 0.558 | 11.530 | 0.000 | Sig | 0.403 | 5.965 | 0.000 | Sig | 0.442 | 9.851 | 0.000 | Sig |
| Trust | H3a | $\mapsto$ | eWOM | 0.090 | 1.837 | 0.098 | Not | 0.147 | 2.496 | 0.023 | Sig | 0.363 | 3.769 | 0.000 | Sig |
| | H3b | $\mapsto$ | Willingness to Use AI | 0.045 | 0.872 | 0.38 | Not | 0.532 | 11.662 | 0.000 | Sig | 0.206 | 2.387 | 0.011 | Sig |
| Customer brand engagement | H4a | $\mapsto$ | eWOM | 0.492 | 8.986 | 0.000 | Sig | 0.450 | 8.075 | 0.000 | Sig | 0.183 | 2.169 | 0.023 | Sig |
| | H4b | $\mapsto$ | Willingness to Use AI | 0.419 | 6.032 | 0.000 | Sig | 0.075 | 1.136 | 0.238 | Not | 0.212 | 2.822 | 0.003 | Sig |
| eWOM | H5 | $\mapsto$ | Willingness to Use AI | 0.232 | 3.220 | 0.006 | Sig | 0.018 | 0.0353 | 0.000 | Not | 0.312 | 5.794 | 0.000 | Sig |

Note: *p*—statistical significance ($p < 0.05$).

AI system quality

Hypotheses H2a and H2b, which concern the influence of AI system quality on trust and brand engagement, have been confirmed in all industries, given the high Beta values (β = 0.447, β = 0.227, β = 0.446) and low *p*-values (*p* = 0.000, *p* = 0.012, *p* = 0.000), indicating the crucial importance of quality AI systems.

Trust

Hypothesis H3a regarding the influence of trust on eWOM has been confirmed in the airline and hotel industries (β = 0.147 and β = 0.363, with *p*-values p = 0.023 and *p* = 0.000), but not in the restaurant industry (β = 0.090; *p* = 0.098). Similarly, H3b, which examines the willingness to use AI, is confirmed in the airline (β = 0.532; *p* = 0.000) and hotel industries (β = 0.206; *p* = 0.011) but not in the restaurant industry (β = 0.045; *p* = 0.380).

Customer brand engagement

Brand engagement has shown a significant impact on eWOM (H4a) and willingness to use AI (H4b) in most sectors. Hypothesis H4a is confirmed in all industries (β = 0.492, β = 0.450, β = 0.183, with *p*-values of *p* = 0.000, *p* = 0.000, *p* = 0.023), while H4b is confirmed in the restaurant (β = 0.419; *p* = 0.000) and hotel industries (β = 0.212; *p* = 0.003), but not in the airline industry (β = 0.075; *p* = 0.238).

eWOM

Hypothesis H5, examining the impact of eWOM on willingness to use AI, is confirmed in the restaurant (β = 0.232; *p* = 0.006) and hotel industries (β = 0.312; *p* = 0.000), but not in the airline industry (β = 0.018; *p* = 0.000), suggesting a lower dependency on consumer decisions in this industry based on the opinions of others.

*4.3. Mediation Role of eWOM on Willingness to Use AI in Service Industry*

Table 7 provides an analysis of the mediating role of eWOM on the willingness to use artificial intelligence in the service industry, investigating hypotheses (H6a and H6b) under different conditions. Findings indicate variability in the mediating role of eWOM between trust and willingness to use AI, as well as customer brand engagement and willingness to use AI. Where significant, eWOM shows a moderate-to-strong impact on increasing readiness to use artificial intelligence in the service industry. The consistency of these effects varies, indicating the need for further research to understand under which conditions eWOM exerts a more significant influence.

In the restaurant sector, our analysis revealed that the pathway from trust through electronic word-of-mouth (eWOM) to the willingness to utilize artificial intelligence (AI) was not statistically significant (*t* = 1.569; *p* = 0.117). However, the pathway from customer brand engagement (CBE) through eWOM to willingness to use AI exhibited partial mediation (*t* = 3.055; *p* = 0.002), indicating that CBE may influence customers' readiness to adopt AI through eWOM in this sector. For the airline industry, the examined pathways, including those from trust and CBE through eWOM to willingness to use AI, did not yield statistically significant results. This suggests an absence of a discernible link between these variables in influencing AI adoption within the sector. Conversely, in the hotel industry, both pathways from trust and CBE through eWOM to willingness to use AI demonstrated partial mediation, with statistical significance noted (trust: *t* = 2.722, *p* = 0.007; CBE: *t* = 2.003, *p* = 0.046). This indicates that eWOM may act as a partial mediator in the relationship between trust or CBE and the willingness to engage with AI technologies within the hospitality sector. These findings highlight that while eWOM may serve as a partial mediator between trust or CBE and AI adoption willingness in the hospitality industry, the impact varies significantly across different service sectors.

**Table 7.** Mediation role of eWOM on willingness to use AI in hospitality.

| Service | Mediation Path | Regression | *t* | *p* | Percentile Bootstrap 95% Confidential Interval | | Hypothesis | Results |
| --- | --- | --- | --- | --- | --- | --- | --- | --- |
| | | | | | Upper | Lower | | |
| Restaurant | T –> eWOM –> WU | 0.021 | 1.569 | 0.117 | 0.056 | 0.002 | H6a | No relationship |
| Restaurant | CBE –> eWOM –> WU | 0.116 | 3.055 | 0.002 | 0.193 | 0.051 | H6b | Partial |
| Airline | T –> eWOM –> WU | 0.003 | 0.320 | 0.749 | 0.022 | −0.011 | H6a | No relationship |
| Airline | CBE –> eWOM –> WU | 0.008 | 0.359 | 0.720 | 0.047 | −0.040 | H6b | No relationship |
| Hotel | T –> eWOM –> WU | 0.113 | 2.722 | 0.007 | 0.204 | 0.041 | H6a | Partial |
| Hotel | CBE –> eWOM –> WU | 0.057 | 2.003 | 0.046 | 0.124 | 0.014 | H6b | Partial |

Note: T = trust; eWOM = electronic Word-of-Mouth; CBE = customer brand engagement; WU = willingness to use AI; *p*—statistical significance ($p < 0.05$); *t*—*t* statistic value; path = –>.

## 5. Discussion

Building upon the insights garnered from our research on AI in the hospitality industry, there is a clear demonstration of how AI influences customer trust, engagement, and behavior across the restaurant, airline, and hotel sectors. Our findings about the relationship between AI Information quality and trust, particularly noted in the restaurant and airline industries, highlight the crucial role of high-quality information for fostering trust. This aligns with the research by Jiang and Lee [117] in the healthcare sector, where information accuracy was found to significantly influence patient trust in AI-powered diagnostic tools, underscoring the universal importance of information quality across different service sectors. Further, the direct effect of AI system quality on customer brand engagement (CBE) across all examined sectors resonates with Morosan and DeFranco's [118] findings in the hotel booking context, where AI's interactive features, which provided tailored responses, significantly boosted customer engagement. This suggests that the quality of AI systems can decisively enhance how customers perceive and interact with brands, emphasizing the need for robust AI system development to ensure positive customer experiences. Our study also points to the significant role of CBE in promoting eWOM, especially in the restaurant and hotel sectors. This relationship implies that customers who are more engaged with a brand are likely to share their positive experiences, thus acting as brand advocates. This finding is paralleled in the work of Zhou et al. [119], who noted that high customer engagement leads to increased sharing of positive experiences on social platforms, enhancing the brand's visibility and reputation. However, the impact of CBE on the willingness to use AI showing variability across sectors, with a noted lack of a significant effect in the airline industry, invites further investigation. Kelly et al. [120] have highlighted how perceived utility and ease of use of AI can vary significantly across different contexts, which could explain the varying levels of AI adoption willingness among sectors. This variability suggests that industry-specific factors might influence how AI technologies are perceived and accepted by customers. Additionally, the mediating role of eWOM between trust or CBE and willingness to use AI in the hotel sector, as opposed to its non-significant mediation in restaurants or airlines, illustrates the complex dynamics of customer behavior in different service settings. Sann et al. [121] observed a similar pattern in high-engagement service settings like hotels, where personal stakes and shared experiences make eWOM a more potent influencer. Our research contributes a deep understanding of AI's role in shaping customer dynamics within the hospitality industry. It not only aligns with but also expands upon the existing literature by showcasing how different aspects of AI implementation from system quality to information accuracy play pivotal roles in influencing customer trust, engagement, and, ultimately, their willingness to embrace AI-driven services. These insights are invaluable for industry stakeholders aiming to leverage AI to enhance customer satisfaction and brand loyalty.

In addressing the objectives set out for this research, we successfully explored how AI devices influence tourists' cognitive, emotional, and behavioral responses in Iran, utilizing

the SOR model. This analysis highlights the complex dynamics between AI technologies and tourists, particularly focusing on how cultural context and technology readiness shape perceptions of AI. Our findings offer insights into the integration challenges and opportunities of AI in developing countries, particularly Iran, contributing valuable knowledge to academia, tourism stakeholders, and policymakers. This study enriches the understanding of AI in global tourism and highlights the unique Iranian context. We also identified a gap in empirical research regarding AI's impact on sustainable business practices. The existing literature often assumes theoretical benefits without empirical support. Our study strengthens these theoretical foundations, aiming to facilitate future global research that could empirically validate AI's impact on sustainability in tourism. Throughout the study, certain assertions were linked to sustainable hospitality practices. These claims were assessed through the lens of sustainability by examining how AI integration influences customer trust, engagement, and willingness to adopt new technologies, which are essential components of sustainable business operations. However, it is crucial to note that while these assessments contribute to our understanding of sustainable practices within the hospitality industry, they do not serve as definitive proof of sustainability [122]. Instead, they complement existing theories and provide valuable insights into how AI can potentially support sustainable endeavors in hospitality. In the study by Dalipi et al. [123], there is also an emphasis on the insufficient attention given to sustainability in tourism and hospitality research. Their findings suggest a need for more intensive research in this area to better understand and address the challenges of sustainable business practices. These findings are consistent with the conclusions of our research, which highlights the importance of further investigation and focus on sustainability in the context of artificial intelligence application in the hospitality industry. They also emphasize the importance of standardizing data and performance assessment metrics to ensure the reliability and efficiency of AI systems in supporting sustainable practices in hospitality. These similar conclusions point to a consistency in the challenges and needs arising in the field of AI application in tourism and hospitality, underscoring the importance of further collaboration and research in this domain.

## 6. Conclusions

Our study has provided significant insights into the influence of artificial intelligence (AI) on customer behaviors and perceptions across the restaurant, airline, and hotel sectors within the hospitality industry. By focusing on various AI-related factors, such as information quality and system quality, and their impact on trust, customer brand engagement eWOM, and the willingness to use AI, this research highlights the potential of AI to enhance customer interaction and operational efficiencies, which are crucial for sustainable business practices. Notably, this study underlines the importance of high-quality AI systems that can enhance trust and customer engagement, key factors that drive the success of businesses in the hospitality sector. Trust in AI technologies is shown to be a cornerstone for customer acceptance, influencing their behavior positively which, in turn, could lead to increased sustainability in business operations through reduced resource wastage and improved service efficiency. Furthermore, the differential impacts observed across sectors suggest that AI integration strategies should be tailored to the unique needs and characteristics of each sector to optimize outcomes. For instance, while AI significantly affects customer engagement in restaurants and hotels, its impact varies in the airline sector, indicating the need for sector-specific approaches in implementing AI technologies. Additionally, this research points to the need for further empirical studies to explore the direct impacts of AI on sustainable business practices. Currently, much of the literature makes theoretical assumptions about these impacts without substantial empirical evidence. This study, therefore, sets the stage for future research that could provide the needed empirical backing to these theoretical claims, enhancing our understanding of AI's role in promoting sustainability within the hospitality industry.

## 6.1. Theoretical Implications

This research enhances our understanding of customer behaviors and perceptions in the hospitality industry, especially regarding AI technologies. It supports and expands existing theories on trust, brand engagement, and electronic word-of-mouth in the AI-driven services context. The variations across different sectors emphasize the need for sector-specific models to capture customer responses accurately. Trust is crucial for AI adoption in the hospitality industry, emphasizing the need for businesses to focus on building and maintaining trust in AI technologies. Trust can bridge the gap between AI information quality and customer engagement. Customer brand engagement remains significant in shaping eWOM and willingness to use AI across sectors. Engaging customers actively with brands is vital for generating positive eWOM and fostering AI adoption readiness. This study reveals eWOM's mediating role between stimulus and organism elements, particularly in the restaurant and hotel sectors. It acts as a conduit through which customer experiences and perceptions influence their willingness to adopt AI-driven services.

Building on the initial findings, the results of this study aim to bridge existing gaps and lay a solid foundation for a comprehensive theory of sustainability within the hospitality sector. By highlighting how artificial intelligence can significantly impact trust, engagement, and word-of-mouth promotion, the research underscores AI's potential not only to enhance operational efficiencies but also to further sustainability objectives, such as reducing waste, boosting customer satisfaction, and promoting environmentally responsible practices. In doing so, this investigation not only fills current knowledge gaps but also proposes a robust basis for theorizing about sustainability in hospitality. This paves the way for future research to explore sustainable AI applications in greater depth, stressing the importance of additional empirical studies to validate these theoretical propositions and to extend the understanding of sustainability's role and implementation in the sector.

## 6.2. Practical Implications

This study's findings provide useful information for businesses in the hospitality, aviation, and restaurant industries. First, they stress the significance of giving AI information quality priority. For the purpose of fostering consumer interaction and increasing trust, it is essential to guarantee the trustworthiness and correctness of AI-generated information. The outcomes also highlight the importance of trust-building strategies. Building confidence in AI-driven services is crucial, and firms should customize their trust-building tactics to cater to the unique needs and expectations of their target markets. Promoting active customer interaction with brands can boost positive eWOM and boost consumers' willingness to adopt AI tools. However, because the effects of variables like trust, brand engagement, and eWOM can differ, methods might need to be tailored for each sector.

Prioritizing AI information quality is crucial for organizations in the hotel sector since it has a big impact on client confidence. AI systems that deliver high-quality information and services promote client trust, which, in turn, increases satisfaction and loyalty. Businesses should prioritize honesty, dependability, and consistency in order to accomplish this. Since it has a big impact on eWOM and the desire to employ AI, active engagement with customers' brands continues to be a key strategy in the restaurant and hotel industries. Customized engagement methods can increase consumer adoption of AI and brand loyalty. Recognizing sector-specific differences is essential, as strategies effective in one sector may not yield the same results in another. Therefore, a nuanced approach to AI adoption and customer engagement is recommended, taking into account the unique characteristics of each sector.

Incorporating the significance of sustainability assumptions into this context, this study's findings underscore the potential of AI to serve as a pivotal tool in achieving sustainable practices within the hospitality sector. Beyond enhancing operational efficiencies and customer satisfaction, AI's role in promoting environmentally friendly practices and reducing waste underscores its value in contributing to sustainability. By prioritizing AI information quality and fostering trust through accurate and reliable AI-generated content,

businesses can not only improve customer engagement but also align their operations with sustainability goals. This alignment is crucial for building a sustainable future in the hospitality, aviation, and restaurant industries, where the reduction of environmental impact and the promotion of sustainable consumer practices are becoming increasingly important. Furthermore, the emphasis on customized engagement strategies and the nuanced approach to AI adoption highlight the importance of incorporating sustainability into the core business strategy. Tailoring these strategies to reflect sustainability goals can enhance brand loyalty and encourage the adoption of AI tools, furthering the industry's overall sustainability objectives. Recognizing the role of trust, brand engagement, and positive eWOM in this process underscores the interconnectedness of AI adoption and sustainability initiatives. By addressing these elements thoughtfully, businesses can pave the way for a more sustainable future, fulfilling the growing consumer demand for environmentally responsible practices.

*6.3. Limitations*

Despite the useful knowledge this study has provided, it is important to recognize its limits. First, because the research is based on cross-sectional data, establishing causality is difficult. These associations could be investigated longitudinally in future studies. Second, because this study only looks at three segments of the hospitality business, the results may not apply to other sectors or geographical areas. Last but not least, the analysis is based on self-reported data, which could be biased. While this study's comparative analysis approach is valuable, it also underscores the inherent variability across sectors. This variability, which led to some differences in the significance of AI-related factors, may be influenced by sector-specific dynamics not fully explored in this study. This research was conducted in the context of Iran, which has unique cultural and economic characteristics. Findings may not be directly transferable to other regions or countries with different cultural norms and economic conditions. This study focused on specific AI-related factors, such as AI information quality and AI system quality. Other AI-related variables not included in this research may also impact customer perceptions and behaviors. The study's cross-sectional design captures a snapshot in time. Longitudinal research could provide deeper insights into the evolving dynamics of AI adoption and customer engagement in the hospitality industry. The reliance on survey data may introduce response bias or social desirability bias, impacting the accuracy of responses. This research on the application of AI technology in the tourism sector, with a specific focus on Iran, not only yields crucial insights for the current context but also lays a solid foundation for future endeavors and developmental aspirations in the field. The significance of this research extends beyond the Iranian example, as the obtained results provide a framework for extrapolation to other developing countries.

In acknowledging the limitations of this study, it is important to recognize the paucity of research on the intersection of artificial intelligence, sustainability, and the hospitality sector. This scarcity of studies underscores the pioneering nature of our investigation but also highlights significant gaps in our understanding. One of the critical challenges in asserting sustainability within this context is the lack of long-term data, which hampers our ability to make definitive claims about the sustainable impact of AI applications in the hospitality industry. The nascent stage of AI technology's integration into predictive models for sustainability further complicates the empirical validation of its benefits. Additionally, the current research focuses primarily on exploring tourists' willingness to use AI within the hospitality industry, without directly incorporating sustainability variables into the prediction models. This decision was guided by the immediate availability of relevant data and the initial scope of inquiry, which aimed to establish foundational insights into consumer attitudes towards AI. However, this approach delineates a clear boundary around this study's conclusions regarding AI's role in enhancing sustainable practices within the sector. Future research could address these limitations by incorporating sustainability variables into the analysis, thereby providing a more holistic understanding of

AI's potential to contribute to sustainable development in hospitality. Such studies would benefit significantly from longitudinal data collection, capturing the evolving dynamics of AI adoption and its sustainability implications over time. By expanding the scope of research to include these elements, subsequent investigations can build on the groundwork laid by this study, offering more nuanced insights into the symbiotic relationship between AI technology, tourist behavior, and sustainability in the hospitality industry.

### 6.4. Future Research

By exploring the intricacies of AI adoption within Iran's tourism industry, this study opens avenues for comparative analyses with other nations, fostering a deeper understanding of how cultural, economic, and technological factors uniquely influence the integration of AI in diverse global settings. As a pioneering study in the application of AI in the tourism sector, this research sets the stage for future investigations that can delve into similar contexts worldwide. The implications of the findings suggest that understanding the challenges and opportunities specific to developing nations is paramount in devising effective strategies for AI implementation. By extrapolating lessons from the Iranian case, researchers can inform and guide future studies, contributing to the broader discourse on the global application of AI in the tourism industry. Moreover, the research underscores the need for continued exploration of AI applications in diverse cultural and economic landscapes, advancing our comprehension of the nuanced dynamics associated with technology adoption. This knowledge becomes instrumental for policymakers, industry stakeholders, and researchers alike, guiding the formulation of strategies that align with the unique characteristics of developing nations. As the global tourism landscape continues to evolve, this research offers a valuable roadmap for navigating the complexities of AI integration, ensuring that advancements in technology are inclusive, culturally sensitive, and responsive to the specific needs of diverse societies.

Incorporating the acknowledgment that these results are deemed a vital starting point for any future endeavors aiming to forecast sustainability within the tourism industry, it becomes clear that they are not only foundational but indeed essential. This study's outcomes, while offering insights into the current state of AI adoption and its perceived benefits within Iran's tourism sector, serve as a crucial baseline from which to evolve our understanding of how artificial intelligence can be harnessed to meet sustainability objectives across diverse global contexts. This initial exploration into the intricacies of AI implementation in a specific regional setting underscores the importance of such foundational research as a precursor to more detailed sustainability predictions in the tourism industry. As we consider the path forward, the necessity of building upon these initial findings is evident. Future research must leverage this foundational knowledge to delve into the complex interplay between AI technologies and sustainability goals. This involves not only expanding the scope of investigation geographically and contextually but also deepening the analysis to include specific sustainability indicators and their relation to AI adoption in the tourism sector. The development of predictive models that can accurately forecast the sustainability impacts of AI, based on this study's results, will be instrumental in moving the field forward.

Moreover, the establishment of a comprehensive theoretical and empirical basis for understanding the potential of AI to contribute to sustainable tourism practices is imperative. This will necessitate a multi-faceted research approach that includes longitudinal studies to assess long-term impacts, comparative analyses across different cultural and economic settings, and the formulation of practical strategies for AI integration that are mindful of sustainability objectives. These endeavors will ensure that the pioneering work undertaken in this study is not seen as an endpoint but as a crucial stepping stone towards achieving a more sustainable, technologically advanced tourism industry globally. By recognizing the indispensable nature of these initial findings as a base for future sustainability forecasting in tourism, the research community is better positioned to tackle the

challenges and seize the opportunities that lie ahead in integrating AI into sustainable tourism development strategies.

**Author Contributions:** Conceptualization, A.R. and T.G.; methodology, A.R.; software, D.V. (Duško Vujačić); validation, J.B., A.S., J.Đ.B. and D.V. (Dragan Vukolić); formal analysis, M.M.; investigation, M.K.; resources, D.S.; data curation, M.B.; writing—original draft preparation, T.G., A.R. and B.D.D.; writing—review and editing, S.R.R.; visualization, D.V. (Dragan Vukolić); supervision, D.V. (Duško Vujačić) All authors have read and agreed to the published version of the manuscript.

**Funding:** This research received no external funding.

**Institutional Review Board Statement:** Not applicable.

**Informed Consent Statement:** Not applicable.

**Data Availability Statement:** The data presented in this study are available on request from the corresponding author.

**Conflicts of Interest:** The authors declare no conflicts of interest.

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
