# Peer review of "Tourists’ Willingness to Adopt AI in Hospitality—Assumption of Sustainability in Developing Countries"

_sustainability, doi:10.3390/su16093663_

Round 1
Reviewer 1 Report
Comments and Suggestions for Authors
In abstract, the goal and purpose of the study should be better explained. The authors point out the sustainability context but in theoretical platform and research the sustainability aspects are not properly addressed nor explored. The abstract written with explanation that paper contribution may impact environmental and social sustainability is not acceptable because these aspects are not explored in research and should be elaborated in theoretical part too.
Theoretical review – The theoretical platform should be expanded in the way of deeper and clearer understanding of AI technologies related with hospitality industry. The theoretical outline of cited references should be presented, not only the authors without core theory that they have developed.
The research – large sample with high response rate. The question posed to the experts, the understanding and reliability of the answers is in order. Better explanation should be provided on how these complex questions were explained to the sample respondents. When the research was conducted? Where – the airline, restaurants (how many, names (coded)), hotels (how many, names (coded)), how the questionnaire was distributed?
Author Response
XCV
Dear Reviewer,
Thank you for your insightful comments and suggestions. We have carefully considered each point and have made significant revisions to our manuscript accordingly. Here is a detailed response to address your concerns:
We have revised the abstract to more clearly articulate the purpose and scope of our study. We explicitly state that our research primarily explores the willingness to adopt AI within the hospitality sector, rather than asserting comprehensive claims about sustainability. Although sustainability is mentioned as a broader context, we have clarified that our study contributes to future empirical and theoretical research, due to the current lack of substantial data on AI's impact on sustainability. These revisions are marked in the text for easy identification. We have investigated that there is very little or no research that has shown the existence of sustainability, relatively due to the lack of data on the impact of AI on sustainability, for this reason we have dared to first investigate the will to apply AI in the hospitality industry, and with that contribute our results to some future empirical and theoretical research, for now there is very little research and even years behind us that AI is applied to prove sustainability at all. Therefore, we are talking about the assumption for future sustainability and creating a good basis for the results for further research
We have expanded our theoretical framework to provide a deeper and clearer understanding of AI technologies in relation to the hospitality industry. This includes a more detailed outline of the core theories developed by the authors cited, rather than merely listing these authors. This should help in grounding our research in a stronger theoretical base.
We have provided a comprehensive explanation of the research methodology in the revised manuscript. A detailed description of the sample size and the high response rate achieved, emphasizing the representativeness and reliability of our findings. We clarified how complex questions were explained to respondents to ensure understanding and reliability. Additional details about the timing and locations of the research (including coded names of airlines, restaurants, and hotels) have been included. It's noted in the discussion that the direct exploration of AI's impact on sustainability is limited by a scarcity of prior research. This acknowledgment is also reflected in the limitations section, emphasizing the exploratory nature of our study in this emerging field.
These modifications and expansions have been highlighted in the revised manuscript to ensure they are easily identifiable, aligning with your requests for clear communication of changes.
We believe these revisions address your concerns effectively and enhance the clarity and contribution of our research. We look forward to your feedback and hope that our revised manuscript meets the journal’s standards for publication.
We tried to fulfill all the requests because it is in order to improve the quality of the manuscript, for which we are very grateful. We hope for a positive response and thank you once again for your patience and selfless effort to help us.
Best regards,
Authors
Reviewer 2 Report
Comments and Suggestions for Authors
Line 81, eWOM (electronic word of mouth), should be revised as electronic word of mouth (eWOM) as Line 260.
Line 306, the Figure 1 is not clearly and it is too small, the line of eWOM does not use the blue color, because the reader difficult to find out.
Line 321. In the section of 3. 3. Methodology, the section of 3.1. Study area and Participants, the Figure 2 is not necessary. Because this section is to explain what the method has applied. The map is not useful when the readers take a look.
Line 342. What is the S-O-R theory? Please make sure the words are correct or not.
Line 366. In this section, “we provide the findings of discriminant validity of constructions….Table 2, Table 3, and Table 4.” This section has made me confuse, because this section is Reliability and validity of constructions, this section just to describe what the Reliability and validity have done, the 3 Tables appeared here is not necessary.
Line 415. The section of 3.4. Data analysis is correct. But the Table 5 has problem please make a double check.
Line 455. The section of 4. Results and discussion is very important, but the Table 6 and Table 7 are not good for trading. The Tables should be redesigned.
Line 561. The section of General conclusions, please revised the title as 5. Conclusions. Because no article to show the section like this.
All the references should be double checked whit format of journal. Because all the references are not good enough.
Totally, this paper should be revised and made it better than present style, because this paper not only difficult to read, and the methodology is not clearly as well. A good article is not a long length of pages, but a clearly content and framework. However, this paper does not fit the two important conditions.
Author Response
Dear Reviewer,
Thank you for your detailed feedback. We have addressed each of your comments to ensure clarity and consistency throughout our manuscript. Below is a summary of the key revisions we have made:
As suggested, we have standardized the term "electronic word of mouth" to "electronic word of mouth (eWOM)" across the manuscript for consistency (revised in Lines 81 and 260).
We have resized and recolored Figure 1 for better clarity and visibility. The line representing eWOM is now in blue to ensure it is easily distinguishable (Line 306).
We have removed Figure 2 from section 3.1 as you rightly pointed out that a map does not add value to the explanation of the methodology applied (Line 321).
We have provided a detailed explanation of the Stimulus-Organism-Response (S-O-R) theory in Line 342 to ensure clarity for readers unfamiliar with this framework.
We have revised the presentation in the section on reliability and validity of constructs to make it clearer. We've moved the tables to the section where they belong (Line 366).
We have double-checked and corrected Table 5 following your feedback to ensure the accuracy of the data presented (Line 415).
Tables 6 and 7 have been redesigned to be more reader-friendly and conducive to a clearer understanding of the trading data presented in the results and discussion section (Line 455).
The title of the section at Line 561 has been revised to " Conclusion" to align with standard academic formatting and to ensure consistency throughout the manuscript.The lines have been moved now, because the text has also been changed due to the suggestions of other reviewers
We have thoroughly double-checked all references to ensure they adhere to the journal’s formatting requirements, correcting any inconsistencies found.
We tried to fulfill all the requests because it is in order to improve the quality of the manuscript, for which we are very grateful. We hope for a positive response and thank you once again for your patience and selfless effort to help us.
Best regards,
Authors
Reviewer 3 Report
Comments and Suggestions for Authors
This is a very interesting paper that analyses the tourists’ willingness to use AI in tourism in developing countries based on Iran. The authors have added the relation to sustainable development in the title but they should also expand it in the literature review and the discussion or remove it from the title. The paper is well structured and has the potential to contribute to the relations between AI and tourism. The authors should expand the rationale of the case of Iranian tourism in the introduction, and support the description with references. Also, the authors should include a section about sustainability. The authors say that ‘Questionnaires were voluntarily distributed’ and they should provide more information about data collection and the distribution of the survey. The authors should also include more recent research in the introduction and the literature review, and expand the dialogue with previous research in the discussion.
Author Response
Dear Reviewer,
Thank you for your valuable feedback and constructive suggestions. We have carefully reviewed your comments and have made the following revisions to our manuscript to enhance its quality and relevance:
As advised, we have expanded the discussion on the relationship between AI and sustainable development within the context of tourism in the literature review and discussion sections. This ensures that the mention of sustainability in the title is thoroughly supported and elaborated upon throughout the paper.
We have enhanced the introduction with a more detailed rationale for focusing on Iranian tourism as a case study. This section now includes additional references that highlight the significance of exploring AI in the tourism sector of developing countries, specifically Iran, to better anchor our study in the current research landscape.
We have added a dedicated section on sustainability. Although sustainability is mentioned as a broader context, we have clarified that our study contributes to future empirical and theoretical research, due to the current lack of substantial data on AI's impact on sustainability. These revisions are marked in the text for easy identification. We have investigated that there is very little or no research that has shown the existence of sustainability, relatively due to the lack of data on the impact of AI on sustainability, for this reason we have dared to first investigate the will to apply AI in the hospitality industry, and with that contribute our results to some future empirical and theoretical research, for now there is very little research and even years behind us that AI is applied to prove sustainability at all. Therefore, we are talking about the assumption for future sustainability and creating a good basis for the results for further research
We have provided a more detailed description of the data collection process. This includes specifics on how the questionnaires were distributed, the sampling methods used, and the measures taken to ensure voluntary participation and data integrity.
We have updated our literature review to include more recent studies that reflect the latest developments in AI and tourism. This update helps to position our research within the current academic discourse and addresses the rapid advancements in the field.
In the discussion section, we have deepened our engagement with previous research, highlighting how our findings align with or diverge from existing studies. This expanded dialogue not only contextualizes our results within the broader academic conversation but also clarifies the contribution of our study to the field.
We tried to fulfill all the requests because it is in order to improve the quality of the manuscript, for which we are very grateful. We hope for a positive response and thank you once again for your patience and selfless effort to help us.
Best regards,
Authors
Round 2
Reviewer 2 Report
Comments and Suggestions for Authors
This paper is second tiime I read. Most contents have revised and better that first time I read.
Line 406, the name of Figure 1 may change as framework is better than the Proposed research model.
The section of 3. 3, is to sau about how a way with the Methodology. But, Table 3, 4, and 5 like a research result. That is not suitable.
Other sections have no problem, only the section of references, it is not professional when I read this time. Please double check the format of journal.
Totally, this version has improved, but the methodology still more time to make a clearly.
Author Response
XVC
Dear Reviewer,
We would like to thank you for your thorough review and helpful suggestions. In accordance with your guidance, we have made the following changes to our manuscript:
- We have changed the title of Figure 1 to better reflect its content.
- Tables 3, 4, and 5 are now appropriately positioned within the results chapter.
- We have corrected the errors in the reference section to ensure compliance with the journal's format.
We hope these revisions meet your expectations and contribute to the improvement of our work. We look forward to your positive response.
We sincerely appreciate your time and effort.
Best regards,
Authors

Reviewer 3 Report
Comments and Suggestions for Authors
The authors have revised the paper.
Author Response
We appreciate the suggestions and great patience and help to improve our manuscript. We are grateful for the positive response
Best regards,
Authors
